# How Can We Improve Running Events? The IPA4RE Questionnaire and Its Potential Impact on the Experience of Runners

**DOI:** 10.3390/ijerph20031703

**Published:** 2023-01-17

**Authors:** Jairo León-Quismondo, José Bonal, Pablo Burillo, Álvaro Fernández-Luna

**Affiliations:** Faculty of Sport Sciences, Universidad Europea de Madrid, Calle Tajo, S/N, Villaviciosa de Odón, 28670 Madrid, Spain

**Keywords:** running, sporting events, importance–performance, adherence

## Abstract

The benefits that a positive running experience provides to individuals have been broadly studied by the scientific community, with the finding that running sport events are a facilitating tool to attract more individuals into physical activity. This study focuses on a sport management approach to improve the quality and organizational efficiency of running sports events so that a better experience for runners can be provided. The methods of this research consist of the validation of a new questionnaire called ‘The IPA4RE questionnaire’. Furthermore, an IPA analysis for a specific event was carried out. As our main findings, the most valued aspects perceived by consumers are the event atmosphere, security, event route, and convenience of bib collection and event day information. In conclusion, the IPA4RE can be used as a management tool by running event organizers to make more efficient use of their resources and provide a better experience to participants.

## 1. Introduction

The establishment of the welfare state and the “sport for all” movement has significantly increased the opportunities for sports participation [1]. Enrollment in sport events has been proven to develop higher rates of activity commitment and future exercise intentions for participants [2] while enhancing their regular exercise intentions and ensuring repeat participation intentions when they have a positive event experience [3]. Additionally, the celebration of mass participant sports events has been proven to be beneficial, from organizational and strategic points of view, for elevating the image of a host city in the long term [4] and for the economic impacts in the area [5,6]. Considering all these benefits, running sport events can be considered a great tool for promoting an active society and providing an economic stimulus to host cities.

The rise of physical activity participation has led to the growth of running races worldwide [7]. Popular endurance events have acquired greater international importance in society as a boost for the tourist demands of a host city. Long-distance or endurance running events have spread all around the globe in the last three decades, including a high diversity in the related event distances (5 km, 10 km, half marathon, marathon, or ultra-marathon) and orientation (urban races, trail races, fun races, or internationally oriented competitions). In Europe, the size of the running market is approximately 45–55 million adult runners [8]. In Spain, the number of running events held is over 3300, reaching more than 40,000 participants with a single event [9]. In this sense, Wicker and Hallmann [10] estimated the willingness to pay by European marathon runners at EUR 568 for events in Europe and EUR 1429 when the event takes place in America. These impacts unquestionably determine the continuity and investment of companies in such events.

Having a specific goal (e.g., running sport event) has been proven to be a great motivator for amateur runners, generating higher rates of training and physical activity adherence [11]. Linking this with the research of Alexandris [12], which explains how a better costumer experience has a positive impact on repurchase intention, customer loyalty, and fan engagement, the main idea of the present investigation is to improve the quality of running sport events. Considering this as a key statement, this study follows a sport management approach driven by the following research questions: a) what are the primary management factors in a running event, and b) is the importance–performance analysis a suitable tool for the evaluation of running events?

Helping companies to efficiently manage their resources and facilitating a better understanding of consumer demands would enrich the participant experience in a sport running event. By knowing the key elements, sport managers can tailor the event to a runner’s profile since different profiles of runners, according to their behaviors, have been established [13], as have the differences in runners’ overall and specific expenditures [14]. Despite the diverse profiles of participants, there is evidence that services such as parking availability, security, access, cleanliness, information points, and visible dustbins along the route and at the sports grounds are key in these events [15]. The growing experience of participants in running races and, specifically, in marathons is producing an increasing rivalry among running events; therefore, the results of the present investigation can be used by organizers to adjust their offerings to the existing demands and establish differentiation strategies, reaching more competitive positions and differentiating their events from similar events.

Thus, to contextualize this research within the wider literature, we reviewed the key management factors in sport events that have been identified by different authors in previous research, followed by defining the importance–performance analysis as a tool for the evaluation of sport services or events.

### 1.1. Key Management Factors in Sport Events

When the key factors in management are discussed, it is convenient to consider the original place that these can be identified through qualitative or quantitative research. Regarding qualitative research, methods are generally based on interviews with managers of different entities in which they identify the key elements of their management, necessary skills of executives, or relationships with stakeholders through their personal experience or opinions [16,17,18]. However, the most widespread method is quantitative research, in which the work of authors such as Calabuig-Moreno and his EVENTQUAL questionnaire [19], which evaluates the satisfaction of the attendees of sporting events, should be noted as a previous step to identifying the key aspects of the management of the event. Thus, in this study, aspects such as accessibility (signposting and sitting), tangibles (viewing and atmosphere), staff (training, kindness, and performance), and complementary services (cleanliness and concessions) are evaluated by the spectators of the event and give an indirect idea of the aspects of the event’s management that can be improved. Another remarkable aspect of the participants’ event perceptions is their individual performance, as stated by the use of the multidimensional analysis PSEASD (participant sport event attribute and service delivery), concluding that the link between an individual’s performance and their overall satisfaction and re-participation intentions can be more of a determinant than event service quality [20,21]. Other research aimed at tourism has added to Calabuig-Moreno’s work, where aspects such as the environment (landscape) where the sports activities take place and perceived safety are evaluated [22], and other investigations in which the social impact of the event is evaluated alongside critical management factors [23]. In this regard, it should be noted that to propose any model for evaluating customer satisfaction in sporting events, it is very important to differentiate between participatory sporting events, events with spectators, and mixed events [24], since the inputs received by a manager can vary greatly depending on whether they are given by participants who evaluate the event or spectators. For this reason, it can be affirmed that designing or applying a standard tool to sporting events is a very complicated task, and for this reason, the identification of critical management factors is established through interviews, focus groups, and questionnaires designed using previous research or ad hoc questionnaires.

### 1.2. Theoretical Framework: IPA Analysis

The importance–performance analysis (IPA), originally developed by Martilla and James [25], allows for the simple and effective diagnosis of organizations, offering associated strategies for each service attribute. Following the IPA, each participant was asked about the level of importance of different elements of the event, as well as the level of performance of the same elements. The difference between the level of performance and the level of importance is known as the discrepancy [26], and it is shown in the IPA matrix as an iso-rating or iso-priority crossing line [27]. The IPA matrix allows for linking every service attribute with an appropriate strategy (Figure 1).

Even though this model was originally created as a marketing tool external to the scope of the sports, the professionalization of the sports industry has generated an interest in the IPA, and it has been applied to sports organizations [28,29,30] and sporting events [23]. This way, optimal managerial decisions can be applied. The IPA is an optimal tool for this purpose, since it allows for comparisons with the previous experiences of participants with their present perceptions. Although the IPA model has already been used in other sporting events to evaluate critical management factors [23], there is no scientific evidence about the validated instruments for measuring the importance–performance analysis in mass running events. Therefore, the main objective of this research is the study of the reliability and construct validity of a new questionnaire based on the IPA, specifically for running events, to help future researchers and sport managers. Additionally, using the IPA approach, the secondary objective is to identify the most valued management aspects of a running event by the users.

## 2. Materials and Methods

### 2.1. Sample Profile and Backgraound Item Generation

The study participants were runners in a mass 10 km running event celebrated in Madrid, Spain. A total of 185 amateur runners (21% women and 79% men) with a mean age of 38.56 ± 10.96 years participated in the research. Their participation in running events was 8.34 ± 7.75 events per year, with a dominant presence of 5 km and 10 km event distance types.

The design of test items for the new IPA instrument was based on a conceptual framework of events satisfaction, following a gap analysis conducted by Martilla and James [25]. The conceptual framework was grounded in a review of the scientific literature that dealt with the IPA score across different leisure and tourism areas [28,29,31,32].

### 2.2. Procedure

The draft conceptual framework was completed by two panels of expert sessions. Management (N = 2) and academic experts (N = 4) were recruited to take part in two one-hour sessions. The academic experts were professors and assistant professors who specialized in sport management and who had more than seven years of experience in sports events analysis. On the other hand, the management experts, who had more than five years of experience, were members of sports federations and events management companies. Once the framework was fully elaborated, the first version of the IPA questionnaire for running events was named the IPA4RE test (IPA for running events). Sociodemographic data and 40 scale items, using a Likert 1–5 measurement, were included. Scale items were divided into two dimensions: importance (20) and satisfaction (20), in line with the IPA methodology previously described [25]. In addition, scale items were divided into three categories in both the importance and satisfaction scales: management items (7), day of event items (8), and itinerary items (5).

Based on feedback from experts, the IPA4RE test was modified, revised, and improved to enhance its clarity and face validity. The experts suggested the elimination of 2 scale items due to duplicity of content. Thus, the final version of the IPA4RE questionnaire included 36 items (18 on the importance scale and 18 on the satisfaction scale). After modification, a pilot study was undertaken to examine the content validity from the perspective of the targeted population and to assess the reliability of the developed scales [33]. The questionnaire was written in Spanish (Appendix A). The data were collected two days before the event (at the bib collection fair) and on the day of the event. The researchers used tablets for data collection using the QUICKTAP SURVEY^®^ software (Toronto, Canada). A total of 185 responses were collected and were later measured with Cronbach’s alpha and item-to-total correlations in order to assess reliability, ranging from 0.200 to 0.614 in the importance scale and from 0.235 to 0.667 in the performance scale. Table 1 presents the initial scale reliabilities and item diagnostics. The Cronbach’s alpha values obtained were acceptable, with the exception of two items that were slightly lower than the value of 0.7. Because of this, none of the items initially proposed were eliminated.

### 2.3. Data Analysis

Data were analyzed through an exploratory factor analysis with a Varimax rotation. The exploratory factor analysis results provided useful information regarding the number of factors based on eliminating and/or combining items and dimensions for representing a more valid factor structure [34]. Bartlett’s Test of Sphericity (BTS), which provides information about whether the correlations in the data are strong enough to use a dimension-reduction technique such as factor analysis, and the Kaiser–Meyer–Olkin (KMO) measure of sampling adequacy values were evaluated [35]. Firstly, the current study used the Kaiser criteria to identify a factor that has an eigenvalue greater than or equal to 1 [36]. Secondly, factor loadings had to be at least equal to or greater than 0.40 to be retained. Finally, the identified factors and items should have been theoretically interpretable. All statistical tests were performed using the IBM SPSS 25.0 statistics software (IBM Inc., Chicago, IL, USA). Significance was set at *p* < 0.05.

## 3. Results

### 3.1. Descriptive Statistics

Table 2 exhibits the descriptive statistics. All the 18 importance items had a mean score of 4.05, meaning that the participants considered most of the elements shown in the survey to be important. The average perceived performance was 4.00, meaning that the perceived performance was slightly lower than the average importance, which shows that the participants felt somewhat dissatisfied. The event atmosphere (item 12) was the most important element for the participants, but it was also the one that showed the best performance. The parking area and accessibility (item 7) were the least important attributes, with correspondingly low levels of performance.

The IPA matrix is shown in Figure 2. This allows for a fast and simple interpretation of the results. This is divided by the discrepancy line, which is reflected on the graph with a 45-degree line. The categories that are above the line have negative discrepancy values and are related to dissatisfaction levels. The elements above the line are of greater importance, and these will be the elements to focus on. Below the line, we have several subdivisions (low-priority, possible overkill, and keep up the good work).

There are 11 of the 18 elements that are above the discrepancy line, meaning that their level of importance was higher than the level of performance, which results in levels of dissatisfaction. Therefore, all these elements are in the concentrate area: 3 (previous information about event day: date, place, time, etc.), 4 (problem-solving before event day), 5 (convenience of bib collection fair), 8 (event day indications/information: starting point and toilets), 11 (organization of start: waves), 12 (event atmosphere), 13 (route), 15 (waste and recycling management), 16 (first aid and problem-solving during the event), and 17 (security).

The low-priority items are 1 (registration fee), 2 (website and application), 7 (parking area and accessibility), 9 (speaker, announcements, and other technical aspects), 10 (baggage lorries), and 18 (finisher’s awards).

Item 6 (kit bag) was identified as a possible overkill element, meaning that there is a possible waste of resources associated with it.

Lastly, there were no items identified as keep up the good work.

### 3.2. Exploratory Factor Analysis

Two separate exploratory factorial analyses were made. Dividing importance impacts into two dimensions was intended to decrease the chance of distorted results due to the direction of signs and inconsistency of meaning among items in the same dimension. For the importance dimension, the KMO measure of sampling adequacy value was 0.84 and the BTS was 1034.47 (*p* < 0.01), indicating that the sample was appropriate for conducting a factor analysis. As a result of the factor analysis with the Varimax rotation, five factors were identified, explaining 59.36 % of the variance. All items exceeded the 0.40 load criteria. One factor, consisting of items 1 (price) and 7 (parking), was removed because its loading onto a factor did not have the appropriate theoretical justification. Therefore, four factors with 18 items, explaining 53.51% of the variance, emerged from the exploratory factorial analysis: event general information and characteristics (five items), problem-solving and security (four items), benefits, atmosphere, and extra services (six items), and bib collection and storage organization (three items).

For the performance dimensions, the KMO measure of sampling adequacy value was 0.88 and the BTS was 1169.01 (*p* < 0.01). The KMO value and the BTS value indicated that the sample was appropriate for a factor analysis. As a result of the factor analysis with the Varimax rotation, four factors were identified, explaining 58.90% of the variance. All items exceeded the 0.40 load criteria. The four factors were named as follows: inscription and rewards (five items), information and support before and during the event (seven items), storage and front desk organization (three items), and facilities and venues (three items). All the final factors of the items and the Cronbach’s alpha values for the factors are shown in Table 3 and Table 4.

## 4. Discussion

The IPA has been used as a management tool in different sports industry areas, such as fitness and events [23,28,30]. The obtained findings prove that, in the specific case of running events, the IPA4RE questionnaire is a valid and appropriate tool. In addition to the validation of the IPA4RE questionnaire, some interesting and valuable information was identified from the results of the current investigation regarding the perceptions of the running events by their runners themselves.

First, taking a deeper look at some items located in the IPA matrix area named concentrate here, such as the route (item 13) or event atmosphere (item 12), these items can be compared with previous research that suggested that event organizers should focus on the overall runner experience to obtain willingness to pay [3,10,37]. In this line, the obtained importance of the route (item 13) is linked to one of the main motivations identified for Cuban and Chilean runners, who acknowledged beating their own time and self-improvement as race-joining arguments [38]. Other interesting elements from this research were previous information about the event day (date, place, and time) (item 3) and problem-solving before event day (item 4). Both items align with the needs and high demands of running events customers, as proven in previous research, as the profiles of participants are those of (largely) highly educated, demanding, and affluent people [10,39]. Further, convenience of the bib collection fair (item 5), waste and recycling management (item 15), security (item 17), and first aid and problem-solving during the event (item 16) were identified as important, which shows coherence with the recommendation of focusing on the transport, security, and ecology aspects of sports events for sports administrators, since these items produce a feel-good effect in the event attendees [40]. On the other hand, one of the low-priority items found in the current study results was registration fee (item 1). This proves that, while facing the challenge of an endurance sport event, the registration fee is not a high-priority aspect in comparison with the experience as a whole [10].

Second, regarding the main research objective related to the construct validity of the questionnaire, high correlations were observed between most of the items studied. The factor loading of each item considered low or as providing little information to the factor was considered below 0.40. Hence, as presented in the results section, the only items that met the condition previously exposed were the following: item 1 (registration fee) and 7 (parking area). Thus, the derived factors have finally underlined four constructs for the importance scale and four for the performance scale. Those facts probe the adequacy of the IPA4RE questionnaire to measure key management factors in sports events.

Third, referring to the importance scale, four dimensions emerged, something comparable with previous studies that have focused on other disciplines highlighted the importance of (a) customer service (e.g., problem-solving and security), which is present, for instance, in fitness services research [30]; (b) events environment and rapport (e.g., benefits, atmosphere, and extra services), which are present in customer experience studies [17]; and (c) facilities (e.g., bibs collection and storage organization), which has a key role in sports events scientific literature [23]. In particular, the weight of the problem-solving and security category was highlighted, reaffirming the statement of the feel-good effect that other sports organizers such as FIFA (Federation Internationale de Football Association) describe at their mega-events, which identifies the high importance of security for sport fans when attending any sporting event [40]. Thus, in digging deeper into the ¨security¨ category, the incremental interest experienced can be justified by a post-covid effect where costumers worry more than they did before about the event’s health and safety conditions [41,42].

Fourth, as previously mentioned, regarding the performance scale, four dimensions emerged after the construct validity of the questionnaire, highlighting the relevant role of some dimensions for customer satisfaction such as information and support before/during the event or facilities and venues. Regarding facilities and venues, our results showed coherence with the high importance attributed by sport fans to facilities, in past studies by, for example, Rodríguez et al. [43], who tested the SERVQUAL questionnaire on soccer fans, or Zhang et al. [44], who tested the same questionnaire on hockey fans. Their work underlines the high importance of accessibility and the services provided by the stadium. Similar dimensions to the present research findings are also shown in customer behavioral studies, which measure customer expectations vs customer perceptions (e.g., SERVQUAL or SPORTSERV), obtaining similar results while grouping the most important factors involved [45,46]. In fact, some SERVQUAL dimensions—responsiveness, reliability, assurance, empathy, and tangibles—are intrinsically linked with most of the IPA4RE-detected factors.

Finally, it is interesting to consider the limitations of the present investigation for a better contextualization of the conclusions. For instance, most runners were Spanish citizens, since all the data were collected just one event in Madrid. Therefore, for the current research, the cultural perceptions were not fully contemplated as has been expressed in previous literature (e.g., Ma and Kaplanidou [3] comparing running events in Taiwan and Greece and analyzing different perceptions between both countries).

## 5. Conclusions

In light of the presented findings and discussion, the main conclusions of the present investigation are as follows: (a) The most valued management aspects of an amateur running event are event atmosphere, security, event route, convenience on bib collection, and event day information. Sport event organizers should design strategies that reinforce the appropriate implementation of these relevant aspects, improving the rates of consumer satisfaction, event re-participation, and runner adherence to physical activity. (b) The IPA4RE questionnaire is a suitable tool for evaluating running events. It is highly recommended that it be used to enable a better understanding of all the different running events and the efficient use of the resources by an event organization.

Future lines of research could focus on obtaining a deeper understanding of the factors that inspire individuals to make the decision to start running training and to participate in a running event. In addition, it could also be interesting to study other events in different countries, or the comparison of physical activity adherence, expenditure, and expectations between amateur occasional runners and amateur runners who prepare for a sport event (e.g., a 10 km running event), aiming to comprehend the influence of having a final goal.

## Figures and Tables

**Figure 1 ijerph-20-01703-f001:**
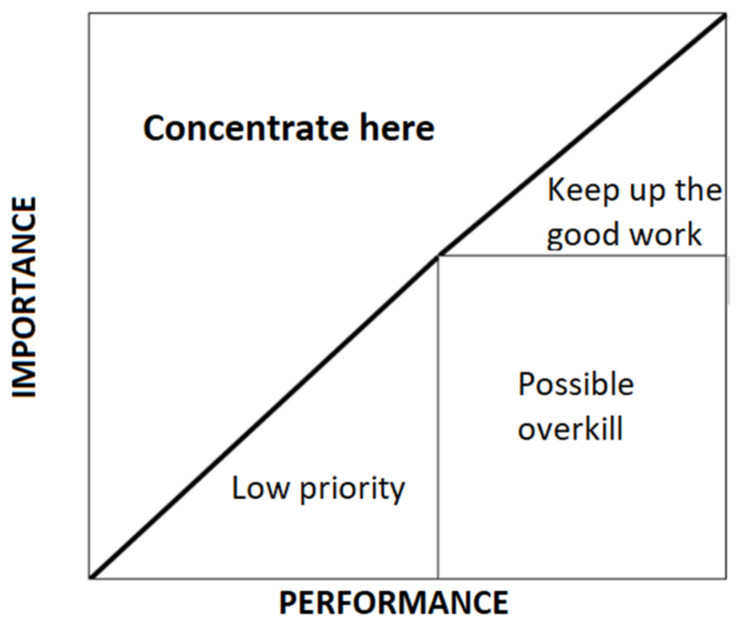
IPA matrix.

**Figure 2 ijerph-20-01703-f002:**
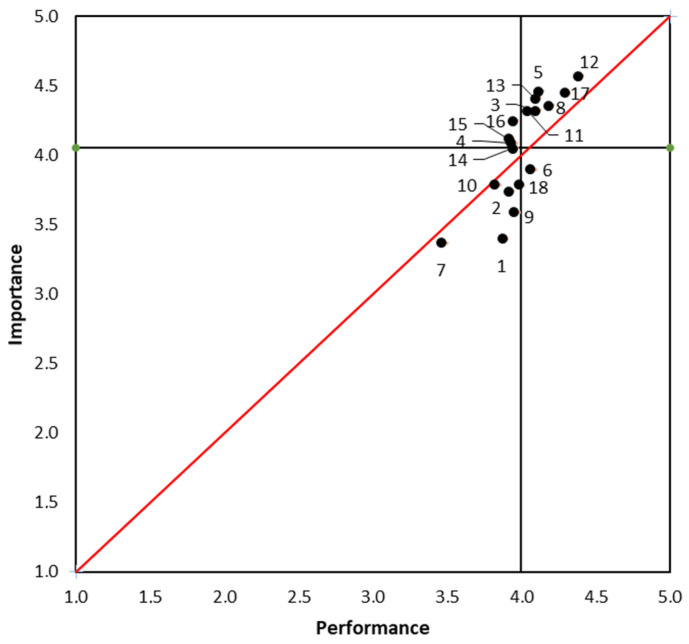
Obtained results as shown in the IPA Matrix.

**Table 1 ijerph-20-01703-t001:** Internal consistency of the IPA pilot study (n = 185).

Scale	Dimensions	Cronbach’s Alpha (α)
Importance scale (18)		0.885
	Management items (6)	0.667
	Day of event items (7)	0.689
	Itinerary items (5)	0.745
Satisfaction scale (18)		0.894
	Management items (6)	0.744
	Day of event items (7)	0.746
	Itinerary items (5)	0.806

**Table 2 ijerph-20-01703-t002:** Means and standard deviations (SDs) for the importance and satisfaction items as perceived by the participants (n = 185).

Items	IMPORTANCE		PERFORMANCE	
	Mean	SD	Mean	SD
1. Registration fee	3.40	1.20	3.87	1.08
2. Website and application	3.74	1.22	3.91	0.93
3. Previous information about event day (date, place, and time)	4.32	0.86	4.04	0.94
4. Problem-solving before event day	4.09	1.06	3.93	0.95
5. Convenience of bib collection fair	4.46	0.82	4.11	1.11
6. Kit bag	3.90	1.03	4.06	0.95
7. Parking area and accessibility	3.37	1.52	3.46	1.20
8. Event day indications/information (starting point and toilets)	4.36	0.84	4.18	0.74
9. Speaker, announcements, and other technical aspects	3.59	1.20	3.95	0.91
10. Baggage lorries	3.79	1.32	3.82	1.16
11. Organization of start (waves)	4.32	0.87	4.09	0.91
12. Event athmosphere	4.57	0.75	4.38	0.79
13. Route	4.41	0.79	4.09	0.99
14. Running supplies	4.05	0.98	3.94	0.90
15. Waste and recycling management	4.12	1.07	3.91	0.97
16. First aid and problem-solving during the event	4.25	0.94	3.94	0.95
17. Security	4.45	0.85	4.29	0.80
18. Finisher’s awards	3.79	1.24	3.98	0.96
Average	4.05		4.00	

**Table 3 ijerph-20-01703-t003:** Factorial analysis: final factor loadings (λ) and Cronbach’s alpha (α) for importance items.

Factors (Items)	λ	α
Event general information and characteristics (5)		0.59
Registration fee	6.63	
Website and application	6.90	
Previous information about event day (date, place, and time)	4.83	
Parking area and accessibility	5.41	
Route	4.84	
Problem-solving and security (4)		0.74
Problem-solving before event day	7.14	
Event day indications/information (starting point and toilets)	5.77	
First aid and problem-solving during the event	6.68	
Security	7.05	
Benefits, atmosphere, and extra services (6)		0.80
Kit bag	5.98	
Speaker, announcements, and other technical aspects	6.48	
Event atmosphere	6.02	
Running supplies	6.67	
Waste and recycling management	5.43	
Finisher’s awards	7.52	
Storage and front desk organization (3)		0.71
Baggage lorries	5.18	
Organization of start (waves)	6.86	
Convenience of bib collection fair	6.45	

**Table 4 ijerph-20-01703-t004:** Factorial analysis: final factor loadings (λ) and Cronbach’s alpha (α) for performance items.

Factors (Items)	λ	α
Inscription and rewards (5)		0.71
Registration fee	6.34	
Website and application	4.31	
Kit bag	7.02	
Finisher’s awards	5.51	
Route	6.90	
Facilities and venues (3)		0.59
Convenience of bib collection fair	7.14	
Event day indications/information (starting point and toilets)	5.77	
Parking area and accessibility	6.68	
Information and support before and during event (7)		0.85
Previous information about event day (date, place, and time)	4.11	
Problem-solving before event day	5.99	
Event atmosphere	4.34	
Running supplies	5.67	
Waste and recycling management	8.18	
First aid and problem-solving during the event	8.81	
Security	6.80	
Storage and front desk organization (3)		0.71
Baggage lorries	8.18	
Organization of start (waves)	6.05	
Speaker, announcements, and other technical aspects	4.49	

## Data Availability

The data presented in this study are available on request from AF.

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
