# Peer review of "How Can We Improve Running Events? The IPA4RE Questionnaire and Its Potential Impact on the Experience of Runners"

_ijerph, 2023, doi:10.3390/ijerph20031703_

Round 1
Reviewer 1 Report
Dear Authors,
You present an interesting and complete study. I really appreciated that you highlight the practical implications for the organizers of running sporting events.
Regarding the 1.1 point, Key Management Factors in Sport Events, it would be interesting to broaden the content: other more up-to-date scales, such as PSEAD (Du et al., 2015; Hyun & Jordan, 2020) could have considered.
In the Material and Methods section, it would be necessary to include what type of race, i.e. the distance (line 122). The profile of the runner differs diametrically depending on the event (5 km course, half marathon, etc.). In addition, more information about the sample is appreciated.
Concerning internal consistency of IPA pilot study (Table 1), although the values are very close to 0.7, there are two that do not exceed this minimum level considered. Clarification would be needed.
Finally, the conclusion section is too concise. It would be positive to include more information about future lines of research in this section.
Author Response
Thank you very much for your feedback

Reviewer 2 Report
León-Quismondo_IPA4RE_ijerph_2022
General comments
Extensive English editing is recommended.
In introduction and aim in the abstract (which is different from the aim in the main text) there has been stated that enrollment in sport events is related to better health but: first: references in the lines of Introduction justifying this approach are incorrect, for example reference number 2 refers to antidepressant adherence and not to running adherence, and second: with the methodology of this study it is impossible to confirm this fact, so I propose to eliminate any reference to better health from the objective of this study.
Furthermore, if we are talking about running events and health, from what the article is not really about, the large amount of literature that relates participation in running events with adverse health effects may not be ignored, for example:
“Abrahim M. MACE in the Race: A Canadian Perspective on Major Adverse Cardiac Events (MACE) During Running. Cureus. 2022 Aug 23;14(8):e28323. doi: 10.7759/cureus.28323. PMID: 36158345; PMCID: PMC9500129.”
Drca N. Wolk A. Jensen-Urstad M. Larsson S.C. Atrial fibrillation is associated with different levels of physical activity levels at different ages in men. Heart. 2014; 100: 1037-1042
Fabian Sanchis-Gomar, Helios Pareja-Galeano, Alejandro Santos-Lozano, Carmen Fiuza-Luces, Nuria Garatachea, Alejandro Lucia, Strenuous Exercise Worse Than Sedentarism?, Journal of the American College of Cardiology, Volume 65, Issue 24,
2015, Pages 2673-2674, ISSN 0735-1097, https://doi.org/10.1016/j.jacc.2015.02.081.
(https://www.sciencedirect.com/science/article/pii/S0735109715019713)
Finke SR, Jänig C, Deschler A, Hanske J, Herff H, Hinkelbein J, Böttiger BW, Schmidbauer W, Schroeder DC. Notfallmedizinische Aspekte bei Laufveranstaltungen [Medical emergencies during running events]. Notf Rett Med. 2021 Dec 2:1-10. German. doi: 10.1007/s10049-021-00959-w. Epub ahead of print. PMID: 34873391; PMCID: PMC8637507.
Due to this incorrect approach of the introduction and the objective of the study, I believe that this article should be rejected in its current form. My recommendation is to adjust the focus of the article to what is really related to the material and methods of it and avoid any reference to health consequences in society.
And after it should be clarified that the main objective of the study is the study of the reliability and construct validity of a new questionnaire (not validity because it is not compared with a gold standar), and the secondary objective is to identified the most valued management aspects of a running event by the users. And according to these objectives rewrite the introduction, the methods (for example in methods in the statistical analysis section, the statistics for the secondary objective are not explained.), the results (the results of the factorial analysis are very confusing: table 3 and table 4 are appropriately named in the text and the Bartlett’s test of Sphericity is not correctly explained), the discussion (line 234: you speak about factors when in reality you are speaking about items and their location in the IPA matrix, confounding the discussion between the outcomes of the IPA matrix and the outcomes of the factorial analysis/ The discussion of the results of the factorial analysis in the 3th paragraph is highly confusing because you do not refers exactly to the dimension detected by your factorial analysis when comparing to the evidence available/ The discussion of the results of the factorial analysis in the 4th paragraphs does not include references in relations to users of running events, the references are related to the perceptions of fans and it is not comparable/The outcomes of the factorial analysis are not discussed from the point of view of the construct validity).
I even suggest to divide the manuscript in two different manuscripts affording the different objectives and making a comparative analysis for the secondary objective, between scales and according to age or sex.
Author Response
Dear reviewer,
We appreciate your feedback. Attending to your comments we have adjusted the focus of the manuscript eradicating the references to health consequences in society and just focusing on the sport management side of the research. As you well pointed out materials and methods focus on this part, therefore we considered a great suggestion to redirect the article towards that direction.
The references that were modified were:
- Mann, C.L.; Rifkin, R.A.; Nabel, E.M.; Thomas, D.C.; Meah, Y.S.; Katz, C.L. Exploring Antidepressant Adherence at a Student-Run Free Mental Health Clinic. Community Ment. Health J. 2019, 55, 57–62.
- Bartlett, J.D.; Close, G.L.; MacLaren, D.P.M.; Gregson, W.; Drust, B.; Morton, J.P. High-intensity interval running is perceived to be more enjoyable than moderate-intensity continuous exercise: Implications for exercise adherence. J. Sports Sci. 2011, 29, 547–553.
- Nowak, P. Ultra Distance Running in View of Health and Amateur Sport. Hum. Mov. 2010, 11, 37–41.
- Xing, X. Validating a running motivation scale in a Chinese culture: The reliability and validity test of the Simplified Chinese Mo-tivations of Marathoners Scales (SCMOMS) 2016, 303–304.
- Crofts, C.; Dickson, G.; Schofield, G.; Funk, D.C. Post-event behavioural intentions of participants in a women-only mass participation sporting event. Int. J. Sport Manag. Mark. 2012, 12, 260.
- Funk, D.C.; Jordan, J.; Ridinger, L.; Kaplanidou, K. Capacity of Mass Participant Sport Events for the Development of Activity Commitment and Future Exercise Intention. Leis. Sci. 2011, 33, 250–268.
- Murphy, N.M.; Bauman, A. Mass Sporting and Physical Activity Events—Are They “Bread and Circuses” or Public Health Interventions to Increase Population Levels of Physical Activity? J. Phys. Act. Heal. 2007, 4, 193–202.
Originally, we read previous literature (Ma & Kaplanidou, 2018) which led us to misunderstand the relationship between running events and their potential as providers for society´s health. However, after reviewing the literature that you pointed out in your comments we have decided to delimit the explanation and terms where running events have really a confirmed influence.
Just in case you would like to check, we have attached the complete reference for Ma & Kaplanidou (2018) we are referring to:
Shang-Chun Ma & Kyriaki Kaplanidou (2018): Effects of Event Service Quality on the Quality of Life and Behavioral Intentions of Recreational Runners, Leisure Sciences, DOI:10.1080/01490400.2018.1448028
Also, we have adjusted the objectives and some of the specific terms you have suggested throughout the document (e.g. items instead of factors in Discussion, previous line 234, current line 262).
Finally, as suggested, we have hired the services of MDPI for the editing of the English for the whole manuscript.
Thank you for taking the time and consideration the review of our research, your advice was used for the improvement of the present manuscript and will be taken into consideration in the preparation of future research.
Round 2
Reviewer 2 Report
León-Quismondo_IPA4RE_ijerph_2022
I thank the authors the effort made to ameliorate the manuscript.
However, I have some major corrections highlighted bellow:
2.2. Procedure
Line 160. You relate that an item-to-total correlations analysis was performed in order to assess reliability, but the outcomes of this analysis for each item has not been included. Is it possible to give at least a range of the values obtained?
Line 170. Please add that: Bartlett's sphericity test provides information about whether the correlations in the data are strong enough to use a dimension-reduction technique such as factor analysis.
Discussion
Lines 244/263. Please change factor to item.
The discussion of the results of the factorial analysis in the 4th paragraphs does not include references in relations to users of running events, the references are related to the perceptions of fans and it is not comparable. If it is possible include more specific references.
The outcomes of the factorial analysis (third and 4th paragraphs) are not discussed from the point of view of the construct validity.
For example, you can add some sentences similar to these:
“Regarding the first objective of the study, the construct validity of the questionnaire, some aspects merit to be discussed. The hight correlations observed between the majority of the items studied and their derived factors have underlined four constructs for the importance scale and four for the performance scale, determined by the dimensions, relevant to key management factors in sports events. (here you add the text you have, because it shows that your underlined constructs fit well with the key factors on the literature).
The factor loading for each item to contribute little information to the factor was considered below 0.4. The items that met the condition previously exposed, were only the following: ……..Those facts probe the adequacy of the IPA4RE questionnaire to measure key management factors in sport events.
Please, add a limitation section at the end of the discussion section.
